# Cross-Sectional and Descriptive Study on the Challenges and Awareness of Hispanic Parents Regarding Their Adolescents’ Mental Health during the COVID-19 Pandemic

**DOI:** 10.3390/healthcare12020279

**Published:** 2024-01-22

**Authors:** Jihye Kim, Robyn Scott

**Affiliations:** 1Department of Secondary and Middle Grades Education, Kennesaw State University, 580 Parliament Garden Way, Kennesaw, GA 30144, USA; 2Department of Educational Leadership, Dalton Public Schools, 1922 W. Brookhaven Circle, Dalton, GA 30721, USA; rscot105@students.kennesaw.edu

**Keywords:** adolescents, COVID-19, Hispanic parents, mental health, mood states, positive feeling

## Abstract

Research has shown that during the COVID-19 pandemic, approximately 20% of children and adolescents in the United States experienced mental health issues that became a significant social concern. However, recent studies have demonstrated that the majority of adolescents maintain positive emotions despite the crisis. This cross-sectional and descriptive study delves into the emotional states of adolescents during the pandemic, considering the viewpoints of both adolescents and their parents, with a specific focus on Hispanic adolescents. Survey results revealed that most adolescents reported positive and happy moods. However, a percentage of adolescents experienced worry, significant changes in feelings, and loneliness as a result of the physical and social isolation associated with virtual learning. Unfortunately, most surveyed Hispanic parents did not adequately recognize their adolescents’ mood changes well. This lack of awareness, caused by factors such as an insufficient understanding about the importance of adolescent mental health, cultural reasons, language barriers, low education, unstable jobs, and more, could lead to missed opportunities for timely mental health interventions. This study seeks to provide a comprehensive discussion on the mental health of adolescents, while also advocating for the emotional wellbeing of Hispanic adolescents.

## 1. Introduction

Adolescence is a developmental stage that is particularly susceptible to being affected by many changes, including those related to mental health. Higher rates of mental health issues among adolescents have become a significant concern in our society, as noted in studies [1,2,3]. The Morbidity and Mortality Weekly Report by Center for Disease Control (2022) reported that approximately 20% of children in the country exhibited mental health problems, but a significant percentage of these children and adolescents with mental health symptoms had not been identified in a timely manner [4]. Consequently, only a few of them have had the opportunity to receive appropriate medical treatment or services [5,6]. Anxiety, depression, and posttraumatic stress are commonly observed among children and adolescents, as highlighted by studies [7,8,9,10,11]. However, these common problems, which are often mild and subtle, can go unnoticed at the initial stages, leaving children and adolescents with mild symptoms unidentified. If left untreated, mild symptoms of anxiety, depression, and posttraumatic stress can develop into mental illness. Such untreated mental illnesses can become challenging to treat and can result in adverse consequences in adulthood, including higher suicide rates, incidents of drug abuse, and other issues [12,13,14].

Extensive research has demonstrated that detecting mental health issues in children and adolescents early on can prevent these issues from extending into adulthood and can facilitate timely treatment and interventions [15,16,17]. However, there are important reasons why symptoms are not identified early. Firstly, it is challenging for children and adolescents to recognize their emotional changes or mental health problems, and mild symptoms of pre-mental health issues are even more difficult to identify. Secondly, parents often lack understanding and do not prioritize their children’s mental health. Many parents tend to be indifferent or reluctant to accept their children’s symptoms [18]. Instead, they perceive their children’s mood states as simple mood changes that reflect adolescents’ developmental stage (i.e., puberty). Distinguishing between puberty-related changes and mental health symptoms becomes challenging for parents, especially when their children perform well at school. Research has demonstrated that children and adolescents with mild symptoms perform academically as well as other students [15,19].

Research has shown that rural and low-income areas, as well as racial and ethnic minorities, have higher rates of mental health problems among adolescents and lower rates of detection [20,21]. With the Hispanic population becoming the second-largest racial or ethnic group in the United States [22], it is important to note that Hispanic individuals with mental health disorders are less likely to use mental health services compared to other groups. Furthermore, parental indifference and lack of recognition of their children’s mental health status appears to be worse among Hispanic parents [23], resulting in a lower likelihood of Hispanic adolescents receiving appropriate and timely medical treatment [24].

The COVID-19 pandemic has had significant effects on the mental health of children and adolescents. The rate of mental health problems among children and adolescents has increased drastically, with approximately 24% of children aged 5–11 and 31% of adolescents aged 12–17 experiencing such problems [25,26,27]. The pandemic has caused adverse health outcomes in the home environment, parental job losses, extended quarantines, and difficulties transitioning to online learning, among other challenges [28,29]. Quarantine restrictions and social distancing requirements in 2020 made detecting mental health problems in children and adolescents even more challenging [30], leading to an increase in unidentified mental health problems. Among the Hispanic population, who have been disproportionately affected by the pandemic due to factors such as language barriers, cultural influences, low education levels, and unstable job situations, the prevalence of depression, suicidal thoughts/ideation, and substance use has increased significantly [31,32,33,34,35]. Despite efforts to promote mental health awareness, Hispanic populations still face significant challenges in accessing support and resources.

While the above studies highlight the negative impact of the pandemic on children’s mental health, recent research has also uncovered a notable resilience and positive emotional wellbeing among adolescents, despite the challenges posed by the pandemic [36,37]. A significant statistical correlation has been found between a happy mood and mental health in youth groups [38]. It indicates that adolescents have displayed a capacity to maintain positive moods which are linked to their overall mental wellbeing.

Along with the literature of adolescents’ mental health, this study seeks to enrich the body of literature on adolescents during the COVID-19 pandemic, as reported by both adolescents and their parents. By doing so, it aims to contribute new insights to the existing dialogue on youth mental health, identifying negative feelings that may signal underlying mental health concerns or instances or, conversely, instances of resilience where adolescents have managed to maintain positive emotional wellbeing amidst the crisis. This cross-sectional and descriptive correlational research specifically targets Hispanic adolescents. This population demands particular attention due to existing studies suggesting a higher prevalence of negative feelings among minority adolescents and their cultural and socio-economic challenges.

Three research questions guided this study:What were the predominant mood states of Hispanic adolescents during the COVID-19 pandemic?Has there been a change in these adolescents’ mood states since the pandemic?How did Hispanic parents perceive their adolescents’ mood states during the COVID-19 pandemic?Have they noticed any changes in their adolescents’ mood since the pandemic?Is there a difference in parents’ perceptions of their adolescents’ recent mood states compared with their perceptions during the pandemic?

## 2. Methods

### 2.1. Samples and Procedure

This cross-sectional and descriptive study was designed as a cross-sectional analysis and adhered to the STROBE guideline for cross-sectional studies, ensuring comprehensive and transparent reporting of our findings. During the 2021–2022 school year, the researcher established a collaborative partnership with a school located in the southeastern region of Georgia, USA. Given the specific goal of the study focusing on Hispanic adolescents, the affiliated school was considered an adequate representation, ensuring that the findings are reflective of Hispanic adolescent mental health during the COVID-19 pandemic. The samples were obtained using convenience sampling due to the practicality and accessibility of the participants within the school setting. Approximately 600 students, aged 12–14, along with their parents from this affiliated school, were invited to participate in the study under the guidance of the school principal. Throughout the academic year, the study collected data from 584 adolescents, with 23 parents voluntarily completing online surveys, without any form of incentive or compensation. To initiate the student survey, the school principal arranged for letters to be sent to parents, outlining the purpose and including a parent consent form. Following this communication, announcements were made to students during school assemblies, accompanied by a minor assent form for their online participation. The parent survey, available in both English and Spanish versions, was distributed in paper form, accompanied by a consent form. The translation of these documents was carried out by qualified native speakers proficient in both Spanish and English. Notably, the student survey was exclusively conducted online and offered in English only.

The student survey yielded 316 female students (54.1%) and a mix of grades, with 162 in grade 6 (27.7%), 191 in grade 7 (32.7%), and 230 in grade 8 (39.4%). Of these students, the majority were of Hispanic heritage (*n* = 423, 72.4%). The parent survey showed that 13 of the 23 parents who completed it (56.5%) used Spanish to communicate with their children. Moreover, more than half of the parents (*n* = 11, 52.4%) had obtained a high school diploma or above, while 13 (59.1%) were working full-time and five (22.7%) were not currently working. The majority of parents were of Hispanic heritage (*n* = 18, 72.4%). More information on the demographics of both groups can be found in Table 1 and Table 2.

### 2.2. Measures

The items used in the parental and adolescent surveys for this cross-sectional study were adapted from established measures such as the KIDSCREEN-10 Index, the anxiety subscale of the Screen for Child Anxiety Related Disorders (SCARED), and the Center for Epidemiological Studies Depression Scale (CES-D). These measures have previously demonstrated good internal consistency, retest reliability, and validity. Specifically, the KIDSCREEN-10 Index has shown a Cronbach’s α of 0.82, ICC of 0.70, and good validity [4,39]. The KIDSCREEN-10 Index maintained a good reliability (α of 0.81) even amidst the COVID 19-related study [29,40]. The SCARED has demonstrated a Cronbach’s α of 0.74–0.93, ICC of 0.70–0.90, and good validity [41,42]. The CES-D has demonstrated robust internal consistency with a Cronbach’s α ranging from 0.84 to 0.87 [43,44] and exhibits strong validity, as evidenced by indices such as RMSEA = 0.049, CFI = 0.99, and Goodness-of-fit = 0.974 [39,45].

The adolescent survey consisted of 15 items, including three questions related to demographics and 12 questions related to their mood state during the pandemic and currently. The parental survey included 24 items, including seven questions related to socioeconomic status and 17 questions related to their children’s mood state during the pandemic and currently. Please refer to Appendix A and Appendix B for the specific questions included in each survey. The responses in both the parental and adolescent surveys excluding the demographic items were reported using a 5-point Likert scale, with options ranging from never (0) to always (4). Both surveys included eight of the same items to compare differences in responses regarding the adolescents’ mood state, such as worry, relaxation, anger, loneliness, and sadness (see Table 3). Further, the online survey was set with a single sign-on feature, ensuring that multiple submissions from the same individual were prevented. To maintain the integrity of the data, suspicious response patterns were reviewed at the stage of the survey data collection and cleaning. Additionally, in order to mitigate common response bias, some survey items were reverse-coded. This approach helped in diversifying the response patterns and reducing the likelihood of patterned answering, thereby enhancing the reliability and validity of the survey responses.

### 2.3. Plan of Analysis

An analysis was conducted to investigate the awareness of Hispanic adolescents and their parents regarding the mood states of the adolescents. The study employed multiple linear regression to analyze the survey data obtained from students, with the dependent variable being Item 11. Item 11 on the adolescent survey sought to determine recent mood changes of the students before and after quarantine, with scores ranging from 0 (much worse) to 4 (much better now). The independent variables, Items 4 to 9 on the adolescent survey and 12 to 14 on the parent survey, were used to gather information about mood states during quarantine, including worry, anger, happiness, crying, sadness, loneliness, focus, use of device, talking with parents, and spending family time, with scores ranging from 0 (never) to 4 (always). Gender was considered as a control variable to examine differences in mood changes between male and female students. Subsequently, a logistic regression was employed to identify the prominent mood among adolescents (*n* = 60) who reported a negative mood change during the quarantine. The dependent variable was Item 11 on the adolescent survey, recorded dichotomously with scores of 0 (about the same) and 1 (worse or much worse). The independent variables included Items, 4 (worried), 5 (happy), 6 (lonely), 7 (cried), and 9 (angry), each scored on a scale from 0 (never) to 4 (always). The quantitative analysis, with statistical significance set at α < 0.05, was carried out using SPSS v.28, an established credibility in conducting statistical analyses for social science research [46].

## 3. Results

### 3.1. Adolescents’ Perception about Mood States Related to COVID-19

The study included 584 adolescents in Grades 6 to 8. The results of the study showed that most of the students felt happy and were aware of how to seek help if they had any mental health concerns. In detail, more than half of the students (*n* = 299; 51.2%) reported feeling happy, despite the pandemic, while only 6% (*n* = 33) of the adolescents reported feeling extremely unhappy. This encouraging result is in line with recent research that uncovered positive emotional wellbeing among adolescents, even in the face of the challenges posed by the pandemic [36]. Additionally, 88.7% (*n* = 518) of the participants reported that they were able to focus well on their tasks, irrespective of the pandemic. The majority of the students (*n* = 415; 71.1%) reported spending time with their families, friends, or other groups like clubs or religious organizations during the pandemic. (See Figure 1 for more details).

While the study unveiled a multitude of positive findings, indicating that a significant majority (approximately 60%) of adolescents experienced happiness and positivity amid the ongoing pandemic (see Figure 1), around 40% of the adolescents felt worried, angry, and lonely and often cried during the ongoing pandemic, in response to negative mood state items. More than half of the adolescents (*n* = 367; 62.8%) found it more challenging to do their schoolwork and tended to spend more time using their smartphones than before the pandemic started. Notably, 62.5% (*n* = 356) of the adolescents did not talk to their parents about their feelings, although 70.7% (*n* = 413) knew where to seek help if they had mental health concerns. Furthermore, 45.5% (*n* = 266) felt worried during the pandemic, while 42.1% (*n* = 246) reported getting angry often, and 46.7% (*n* = 273) felt lonely. Another 32.4% (*n* = 189) reported that they often cried during the pandemic. When asked how they feel currently compared to during the pandemic, 45.4% (*n* = 265) responded that they felt better, while 10.3% (*n* = 60) felt worse. A total of 73.3% (*n* = 428) spent more time using electronic devices during the pandemic. The noticeable finding is that more than half of the adolescents felt the same (*n* = 257) and 10% (*n* = 60) did not feel better after three years of COVID-19 quarantine. This discovery illustrates that the mood state of adolescents remains stable despite the challenges posed by the pandemic. Of those who felt worse, female adolescents (*n* = 37; 61.7%) responded feeling worse than male adolescents (see Figure 2 for more details).

### 3.2. Parental Awareness of Adolescents’ Mood States

The response rate from the 23 parents was relatively lower than that of the adolescents (as shown in Figure 3). According to the parents, their adolescents were happy (*n* = 17; 74%) and focused well at school (*n* = 19; 82.6%). They rarely or never cried (*n* = 19; 82.6%), felt angry, worried, or lonely (*n* = 21; 91.3%). Most of the parents believed that their adolescents’ mood state had not changed (*n* = 16; 69.6%) or was about the same or better (*n* = 21; 91.3%) while in COVID-19 quarantine. 69.6% (*n* = 16) of parents noticed no depressive symptoms. Eighteen parents (78.3%) thought that their adolescents spent time with family members, friends, or other groups (e.g., clubs or religious groups), but the adolescents (62.5%; *n* = 356) responded that they did not discuss their feelings with their parents.

### 3.3. Factors Associated with Adolescents’ Mood Change

The regression analysis (Table 4) showed that several factors were significantly associated with the adolescents’ mood state change. These included signs of crying, focus, device use, talk, and family time. Specifically, adolescents who spent more time with their families (*β* = 0.108, *t*(10) = 3.009, *p* = 0.003) and talked with their parents were more likely to feel good or better now compared to during the COVID-19 pandemic (*β* = 0.117, *t*(10) = 2.798, *p* = 0.003) and focus better on what they were doing (*β* = 0.111, *t*(10) = 2.635, *p* = 0.009). In other words, having supportive and open communication with parents or spending quality time with family may have a positive impact on the adolescents’ mood state and ability to focus. Conversely, factors such as increased device use, signs of crying, and lack of communication with parents were associated with a more negative mood state. The overall model had a significant effect, with a low-to-moderate explanatory power ($R^{2}$ = 0.106).

A further analysis revealed that the strongest factor related to adolescents’ current worse mood change was the frequency of crying during the quarantine period, *β* = −0.09, Wald (1) = 0.33, *p* < 0.001. Previous research has suggested that crying without an apparent reason is a common symptom associated with depression and anxiety [47] (see Table 5).

## 4. Discussion

The results of the survey largely confirm the research questions regarding the predominant mood states of Hispanic adolescents during the COVID-19 pandemic. Approximately 60% of Hispanic adolescents in the current study did not experience negative mental health impacts, an interesting finding given the significant lifestyle changes caused by the COVID-19 pandemic. These changes, including fear, uncertainty, loneliness due to physical and social isolation, and virtual learning, have been known to negatively affect children and adolescents’ mental states. However, in this cohort, adolescents predominantly maintained their mood stability and felt positively despite the ongoing crisis, suggesting resilience and adaptability in the face of adversity [48]. Conversely, about 40% of Hispanic adolescents reported feelings of worry, anger, loneliness, or frequent crying during the pandemic. While most negative mood changes among adolescents are not necessarily indicative of severe mental health issues, some reported symptoms may be suggestive of depression or anxiety [49,50].

Notably, the majority of participants (*n* = 356) indicated that they did not communicate their feelings to their parents or caregivers, pointing towards a potential gap in familial communication and support. Without direct comparison data on parents’ perceptions before and during the pandemic, it’s challenging to determine if there is a difference in parents’ perceptions of their adolescents’ recent mood states compared with their perceptions during the pandemic. However, the noted lack of communication about mood states suggests that parents might not have a complete or accurate understanding of any changes that have occurred, emphasizing the need for improved dialogue and awareness of mental health within families. Furthermore, the lack of communication between parents and their adolescents may lead to delayed or insufficient interventions to address mental health concerns [51]. This issue is further compounded for Hispanic families who face challenges related to job insecurity, financial instability, limited knowledge of the healthcare system, language barriers, and cultural factors such as stigma surrounding disabilities. These challenges make it even more difficult for Hispanic parents to prioritize their own mental health, let alone that of their adolescents [18,52,53,54,55].

Regarding the change in adolescents’ negative mood states since the pandemic, the survey only captured a small portion of the data, but it is still significant that the pandemic has had a considerable effect on some of their mental health. This aligns with the general understanding that such unprecedented times can lead to increased emotional stress among adolescents.

Given the study’s findings, it is recommended that practitioners adopt a culturally sensitive approach in engaging with minority adolescent populations, researchers should aim for a more diverse sample in future studies, educators ought to incorporate mental health awareness into their curricula to better support students, and practitioners need to consider the implications of these findings in the development and implementation of adolescent mental health programs.

## 5. Limitations

This cross-sectional study has a primary limitation related to potential selection bias, as the sample of Hispanic adolescents are parents was confined to a single middle school in the southeastern region of Georgia. This limits the generalizability of the results to other minority populations. Additionally, the small number of parents in the sample makes it difficult to compare their perceptions with those of the adolescent participants. Another limitation of this study arises from the utilization of varied wording choices (e.g., Black vs. African American) across the two surveys or inconsistent format of questions (e.g., “how often are you angry” vs. “have you cried often”). While there was no intention to employ different question formats, it is recommended that consistent terminology and question formats be maintained in future studies to enhance coherence and comparability. It is also imperative to engage in longitudinal research to track the progression of mood states over time, elucidating long-term effects and recovery trajectories. Investigating communication barriers between adolescents and their parents or caregivers is crucial in fostering open dialogue regarding mental health issues. Additionally, identifying and promoting resilience factors prevalent among the majority of the study population should be a focal point of subsequent research endeavors.

## 6. Conclusions

While numerous studies have delved into the negative influences of COVID-19 on the mental health of children, recent studies have brought to light a notable resilience and positive emotional wellbeing among adolescents despite the challenges by the pandemic. Researchers have identified a significant relationship between a happy feeling and mental health. The primary objective of this cross-sectional study was to explore the emotional states of adolescents during the COVID-19 pandemic, considering the perspectives of both adolescents and their parents. The outcome of this research demonstrates a compelling finding that the majority of surveyed adolescents and parents reported positive and happy feelings. Although negative emotions were present, their prevalence was relatively minor. This suggests that, overall, adolescents exhibit resilience in managing the emotional impact of external factors such as social distancing, pandemic, and lockdown situations, under their emotional fluctuations expected during their developmental stage.

However, it is crucial to note that some adolescents did express negative moods. This highlights the importance of parental attentiveness to changes in their children’s mood, decreased interest in studying, lower life satisfaction, poorer social skills, lower self-esteem, and increased electronic device usage. Also, open communication between parents and children was encouraged to cope with potential mental health concerns. This insight underscores the significance of parent involvement in supporting adolescents’ emotional wellbeing and resilience during challenging times.

## Figures and Tables

**Figure 1 healthcare-12-00279-f001:**
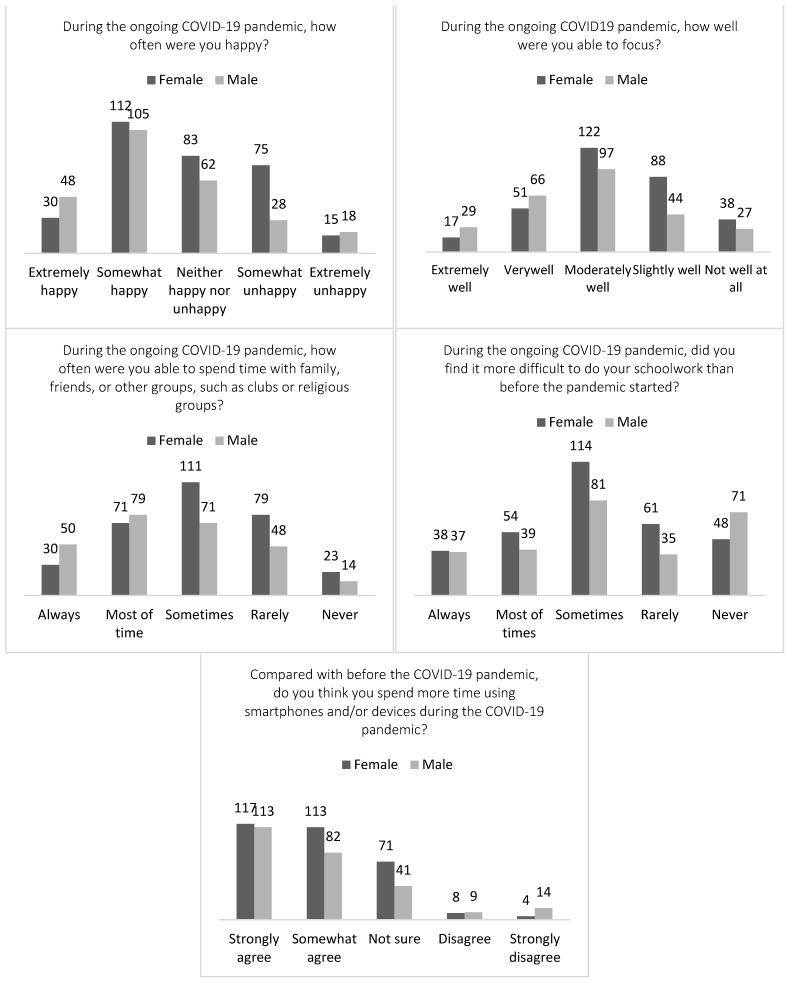
Descriptive Summary of Adolescents’ Responses About Positive Mood States.

**Figure 2 healthcare-12-00279-f002:**
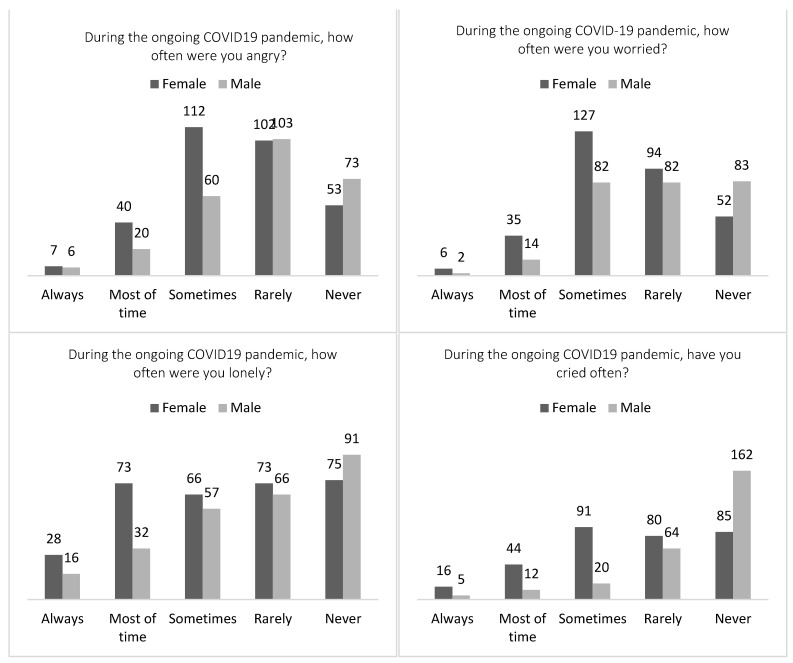
Descriptive Summary of Adolescents’ Responses About Negative Mood States.

**Figure 3 healthcare-12-00279-f003:**
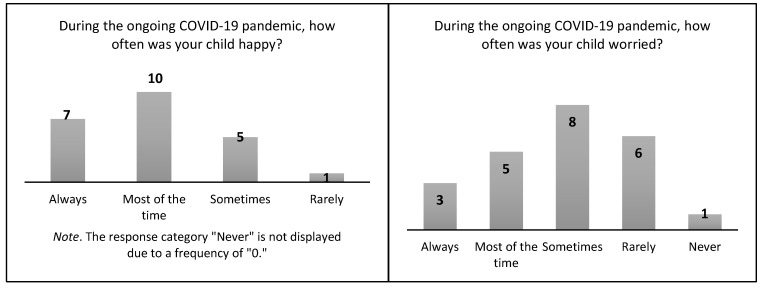
Descriptive Summary of Parents’ Responses About Adolescents’ Mood States.

**Table 1 healthcare-12-00279-t001:** Demographic Summary of Adolescent Survey (*n* = 584).

Category		Value	%
Grade	6	162	27.7
	7	191	32.7
	8	230	39.4
Gender	Female	316	54.1
	Male	263	45.0
Race/Ethnicity	Black	17	2.9
	Hispanic	423	72.4
	Other/Unknown	92	15.8
	White	47	8.0

Note. The discrepancy of frequencies between each category and the total number (*n* = 584) arose from missing values in survey responses.

**Table 2 healthcare-12-00279-t002:** Demographic Summary of Parental Survey (*n* = 23).

Category		Value	%
Most-used language at home	English	10	43.5
	Spanish	13	56.5
Child in special education	Yes	1	4.3
	No	22	95.7
Child in gifted program	Yes	11	50.0
	No	11	50.0
Race/Ethnicity	European American	3	2.9
	Hispanic	18	72.4
	Other/Unknown	1	15.8
Education level	≥high school diploma	11	52.4
	<high school diploma	8	38.1
	Unknown	2	9.5
Employment status	Full time	13	59.1
	Part time	3	13.6
	Not working	5	22.7
	Other/Unknown	1	4.5

Note. The discrepancy of frequencies between each category and the total number (*n* = 23) arose from missing values in survey responses.

**Table 3 healthcare-12-00279-t003:** Common Items in Parental and Adolescent Surveys.

Item on Adolescent Survey	Item on Parental Survey	Item
4	8	During the ongoing COVID-19 pandemic, how often were you/your child worried?
5	9	During the ongoing COVID-19 pandemic, how often were you/your child happy?
9	10	During the ongoing COVID-19 pandemic, how often were you/your child angry?
6	13	During the ongoing COVID-19 pandemic, how often were you/your child lonely?
7	11	During the ongoing COVID-19 pandemic, have you/your child cried often?
8	17	During the ongoing COVID-19 pandemic, how well were you/your child able to focus?
11	19	Compared to the ongoing COVID-19 pandemic and now, how would you/your child the general feeling now?
13	15	Compared with before the COVID-19 pandemic, do you think you/your child spend(s) more time using smartphones and/or devices during the COVID-19 pandemic?

**Table 4 healthcare-12-00279-t004:** Multiple Linear Regression (*n* = 563).

Predictor	B	SE	β	CI 95% Lower	CI 95% Upper	t	df	*p*	VIF
Worried	−0.050	0.047	−0.049	−0.143	0.042	*n* − 1.073	10	0.284	1.263
Happy	0.069	0.040	0.075	−0.010	0.147	1.720	10	0.086	1.169
Angry	0.009	0.048	0.009	−0.086	0.103	0.180	10	0.857	1.415
lonely	0.004	0.028	0.006	−0.052	0.059	0.125	10	0.900	1.462
cried	−0.085	0.045	−0.098	−0.174	0.003	−1.889	10	0.059	1.640
focus	0.111	0.042	0.119	0.028	0.193	2.635	10	0.009 **	1.248
device	0.079	0.040	0.087	0.000	0.158	1.960	10	0.051	1.069
talk	0.117	0.042	0.117	0.035	0.198	2.798	10	0.005 **	1.108
Family time	0.108	0.036	0.128	0.037	0.178	3.009	10	0.003 **	1.208
Gender	0.022	0.090	0.011	−0.153	0.198	0.251	10	0.802	1.210
$R^{2}$		0.106							
*F*		6.542							
Durbin-Watson		1.996							

Note. ** *p* < 0.01; DV: current mood change.

**Table 5 healthcare-12-00279-t005:** Logistic Regression for Children Who Felt Worse Currently (*n* = 63).

Predictor	B	SE	Wald	df	*p*	Exp (B)
Worried	−0.04	0.16	0.06	1	0.800	0.96
Angry	−0.09	0.16	0.33	1	0.560	0.91
Cried	−0.42	0.14	9.61	1	0.001 ***	0.66
Lonely	−0.07	0.1	0.56	1	0.450	0.93
Happy	0.31	0.14	5.17	1	0.020 **	1.36
2 Log likelihood			75.171			
Cox & Snell R Square			0.750			
Nagelkerke R Square			0.103			

Note. *** *p* < 0.001, ** *p* < 0.01; DV: current mood states (dichotomous).

## Data Availability

The data presented in this study are available on request from the corresponding author.

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
