# Peer review of "Cross-Sectional and Descriptive Study on the Challenges and Awareness of Hispanic Parents Regarding Their Adolescents’ Mental Health during the COVID-19 Pandemic"

_healthcare, 2024, doi:10.3390/healthcare12020279_

Round 1
Reviewer 1 Report
Comments and Suggestions for Authors
This is a study of the mental health of Latinx adolescents in one Georgia, USA, middle school as perceived by both the adolescents and their parents conducted in relation to the effects of COVID-19.
The writing is clear and the English is very good. As well, the authors provide references when they are required. On the surface, this appears as a well-done study. However, there are a number of major problems. The majority of these arise from the authors beginning with the pre-conceived notion that Latinx adolescents are likely to have increased mental health difficulties from other students and that their parents are unlikely to notice this because of their assumed concern with academics in contrast to their child’s mental health. Given these assumptions, this is the way the results were interpreted by the authors as they were, although the results of the study don’t support this view presupposed by the authors.
This paper needs to be rewritten in a manner that relates to what actually was found in the study and nothing more. Given their presumption that there are particular and greater mental health difficulties with Latinx adolescents during, and resulting from, COVID-19, the authors want to make conclusions regarding what the school, teachers, and counselors should do concerning the students’ mental health. However, these type of recommendations cannot be made given that the study did not test for these.
The graphs related to adolescents are in blue, wide, and have a shadow, while those of the parents are grey, narrow, and have no shadow. Please be consistent with the width of the bars and whether there is a shadow or not. If the authors want to differentiate between the graphs of adolescents and parents, use only a change in color from blue to grey as the difference.
If the authors check the template for Healthcare, they will notice the following information regarding References: “References must be numbered in order of appearance in the text (including citations in tables and legends) and listed individually at the end of the manuscript.” Based on this requirement, the authors are asked to redo their citations and references in the style preferred by the journal.
Line by line suggested edits.
9 Change “around” to “approximately”.
16 Change “lack of understanding about” to “insufficient understanding regarding”.
23 Add “COVID-19” to the list of Keywords as the first Keyword (as it appears first in the Abstract).
30-33 The statistics quoted are from 2013—ten years ago. Please find current statistics to quote and a peer reviewed references cited to support the statistics from within the last five years.
33-44 None of the research referenced in these citations is from the last five years. Please find additional current references to support the observations.
62 Please provide a definition of “Latinx” and cite a current peer reviewed reference to support this definition.
75-76 Please delete the Bradley et al., 2008 reference as it can have nothing relevant to provide regarding the COVID-19 pandemic.
95-103 These research questions are not well defined. Research questions cannot be compound. As well, they need to be distinct. Change:
“Three research questions guided this study:
1. What was the mood state of Latinx adolescents during the COVID-19 pandemic,
and has there been a change in their mood state since the pandemic?
2. How do Latinx parents perceive their adolescents’ mood state during the COVID-19 pandemic, and have they noticed any changes in their adolescents’ mood since the pandemic?
3. What were the predominant mood states experienced by Latinx adolescents during
the COVID-19 pandemic, and is there a difference in parents’ perceptions of their adolescents’ recent mood changes?”
to
“The following research questions guided this study:
1. What were the predominant mood states of Latinx adolescents during the COVID-19 pandemic
2. How did Latinx parents perceive their adolescents’ mood states during the COVID-19 pandemic
3. Has there been a change in these adolescents’ mood states since the pandemic?
4. Have parents noticed any changes in their adolescents’ mood since the pandemic?
5. Is there a difference in parents’ perceptions of their adolescents’ recent mood states compared with their perceptions during the pandemic?”
106-118 Please state:
1. The relationship of the researchers to the school, the parents, and the adolescents
2. How informed consent was obtained
3. Whether there was an incentive to participate
4. How participants were contacted to be involved in this research
5. How the survey was designed
6. Any exclusion criteria for participation (whether parents were required to have access to the means to complete online surveys during the school day).
110 If the questionnaire was sent home, in what way was this an online survey? Please explain in the text.
120 Change “Language mostly used at home” to “Most-used language at home”. If N=23, why do, Children in the gifted program, Race/Ethnicity, and Employment status add up to N=22? As well, why does Education level have N=21? Please explain in the text.
127-130 As well as these citations, please reference studies that have validated each of the index and scales for use during COVID-19.
140 How do the researchers know that each survey took less than 10 minutes to complete? Please explain in the text.
141 Change “you/your child spend” to “you/your child spend(s)”.
168-168 The graph in the top right corner is the same as the one in the bottom left corner. Only three graphs are needed in this regard, not four.
171-181 By indicating that over 40% of the adolescents had negative emotions during the pandemic, the authors are not highlighting that the majority of the adolescents had positive emotions. This should be what is highlighted, not the negative.
172-181 There is no graph provided for either challenge regarding schoolwork or for smartphone usage. Please include these graphs.
182 To say that more than half of the adolescents did not feel better after three years of COVID-19 quarantine misrepresents the information provided in the bottom left corner graph found in line 187. The majority felt about the same, very few felt worse or much worse. Please indicate that the majority felt either about the same or better than before the pandemic—as the graph demonstrates.
184 Figure 2 does not provide any details regarding the breakdown between female and male adolescents. Please provide graphs that indicate this breakdown.
201-205 The bar graphs of the parents have not been prepared to make them comparable to those of the adolescents in the information they provide. Similar to the adolescents, each bar graph should have five bars. Some have only four, or even three. These bar graphs must be redone to provide comparable information to those of the adolescent graphs.
191 What in the answers provided by the parents makes the authors conclude that parents focused more on their adolescents' academic-related concerns? Please explain in the text how the authors came to this conclusion.
232-234 How were the authors able to come to these conclusions regarding the results that were provided to the questions asked of both the adolescents and the parents? There were no questions asking either the adolescents or the parents if they thought their moods were associated or not with the quarantine. These conclusions cannot be drawn as a result.
240 As per the template provided to the authors, please separate this section into 4. Discussion, and 5. Conclusion.
241-243 Looking over the results of both the adolescents and the parents, their responses were not that different. Consequently, it is unclear if this study’s findings “align with previous research” or not. Furthermore, the studies referenced here are all pre-COVID-19 and are, therefore, irrelevant to questions concerning COVID-19. The authors need to be precise regarding what can actually be concluded and refer only to other COVID-19 studies.
244 The authors says that the change has been “significant”. How does the data from the study demonstrate this? Please explain in the text.
248,-250 Although previous, non-COVID-19-related research may demonstrate this, how did the results of this study demonstrate that the parents prioritized academic support? Please explain in the text. As well, the research needs to be comparable. This means that other studies referenced must be COVID-19-related.
250 Again, the authors have focused on the 40% rather than the 60% who did not experience negative mental health. This skews the results inappropriately.
258 Merely because the adolescents did not communicate their feelings to their parents does not mean that it was “inadequate”. To make this claim, the authors have to provide research to demonstrate that there is a problem with adolescents not communicating their feelings to their parents in the adolescents having sufficient interventions. The authors found half of the adolescents were part of the gifted program. However, the authors did not breakdown the results regarding the responses of the gifted students and their parents compared with the non-gifted. It is possible that the gifted students did not feel the need to communicate with their parents in this regard because they felt they were taking care of their problems themselves. If making claims about the problem of adolescents communicating with the parents, the authors are asked to breakdown their results between those students enrolled in the gifted program and those not enrolled—both for the results of the adolescents and of their parents.
265-275 How do the results obtained demonstrate the importance of Latinx adolescents requiring greater attention and support from schools for their emotional well-being? Please indicate the results that lead the authors to this conclusion. Furthermore, the authors are asked to reference COVID-19 studies. Only one of the ones referenced in the paragraph is relevant to COVID-19 considerations.
291-294 This was a study of adolescents and of parents. It was not a study of the role of schools. The authors can draw no conclusions concerning the role that schools might play in this regard as there was nothing in the study conducted that could provide this information. These recommendations must be eliminated.
294-295 As this study did not consider either teachers or counselors, it cannot provide advice to them. The authors have no information on what teachers or counselors can or cannot do based on the study conducted. These recommendations must be eliminated.
295-298 There is nothing provided in the results of this study to indicate the necessity of interactive communication between parent and school regarding COVID-19-related mental health of adolescents. As well, the paper referenced is from 2011—well before COVID-19 considerations. This recommendation must be eliminated.
298-309 Again, the conclusions drawn here go well beyond the scope of what was determined by the study conducted and the research supporting these conclusions are all pre-COVID-19 by many years. The only type of conclusions the authors can draw are those related to the findings of their study, nothing more.
331 Please check the wording of the question that was on the survey, “During the ongoing Covid-19, how relaxed were lonely?” does not make sense. If this was the wording used on the survey, it must be specified as a limitation of this study.
352 “What grade are your child in?” is incorrect English. Is this what was on the survey? If so, the problem with the English should be noted. The question should have been either “What grade is your child in?” or “What grades are your children in?”.
354 “Is your child in gifted program?” is incorrect English. Is this what was on the survey? If so, the problem with the English should be noted. The question should have been “Is your child in the gifted program”.
355 In Appendix A, line 326, the children were asked if they were “Black” while, in this line, parents were asked if they were “African American”. Please explain in the text why there was a different wording choice for parents—especially because when the data were tabulated there was no mention of “African American”.
Comments on the Quality of English LanguageThe English generally is very good. The few suggested edits are made in the Comments and Suggestions for Authors.
Author Response
Dear reviewer, I sincerely appreciate your thoughtful review and insightful feedback. I have diligently worked to incorporate your comments and enhance the manuscript comprehensively. Below, you will find the responses attached to each comment provided. Thank you once again for your valuable time and consideration!
Response to Review #1 Comments:
First of all, thank you so much for taking the time to review this manuscript. Please find the detailed responses below and the corresponding revisions in track changes in the re-submitted files.
Reviewer’s General Evaluation:
The writing is clear and the English is very good. As well, the authors provide references when they are required. On the surface, this appears as a well-done study. However, there are a number of major problems. The majority of these arise from the authors beginning with the pre-conceived notion that Latinx adolescents are likely to have increased mental health difficulties from other students and that their parents are unlikely to notice this because of their assumed concern with academics in contrast to their child’s mental health. Given these assumptions, this is the way the results were interpreted by the authors as they were, although the results of the study don’t support this view presupposed by the authors. This paper needs to be rewritten in a manner that relates to what actually was found in the study and nothing more. Given their presumption that there are particular and greater mental health difficulties with Latinx adolescents during, and resulting from, COVID-19, the authors want to make conclusions regarding what the school, teachers, and counselors should do concerning the students’ mental health. However, these type of recommendations cannot be made given that the study did not test for these.
Response: Thank you for your feedback. We agree with this comment. I endeavored to interpret the results objectively and removed assumptions about the academic and mental health of Latinx children. Moreover, I have revised the terminology from "Latinx" to "Hispanic" to better reflect the predominant representation of Hispanic samples in the collected data.
The graphs related to adolescents are in blue, wide, and have a shadow, while those of the parents are grey, narrow, and have no shadow. Please be consistent with the width of the bars and whether there is a shadow or not. If the authors want to differentiate between the graphs of adolescents and parents, use only a change in color from blue to grey as the difference.
Response: Thank you for the comment. I have revised the graphs as suggested.
If the authors check the template for Healthcare, they will notice the following information regarding References: “References must be numbered in order of appearance in the text (including citations in tables and legends) and listed individually at the end of the manuscript.” Based on this requirement, the authors are asked to redo their citations and references in the style preferred by the journal.
Response: I have thoroughly revised the references and citations. Thank you for the feedback.
Line by line suggested edits.
9 Change “around” to “approximately”.
Response: I have changed.
16 Change “lack of understanding about” to “insufficient understanding regarding”.
Response: I have changed.
23 Add “COVID-19” to the list of Keywords as the first Keyword (as it appears first in the Abstract).
Response: I have added.
30-33 The statistics quoted are from 2013—ten years ago. Please find current statistics to quote and a peer reviewed references cited to support the statistics from within the last five years.
Response: I have incorporated the latest study into the content.
Bitsko, R. H., Claussen, A. H., Lichstein, J., Black, L. I., Jones, S. E., Danielson, M. L., ... & Meyer, L. N. Mental health surveillance among children—United States, 2013–2019. MMWR supplements, 2022, 71(2), 1.
33-44 None of the research referenced in these citations is from the last five years. Please find additional current references to support the observations.
Response: I have incorporated two latest studies into the content.
Elharake, J. A., Akbar, F., Malik, A. A., Gilliam, W., & Omer, S. B. Mental health impact of COVID-19 among children and college students: A systematic review. Child Psychiatry & Human Development, 2022, 1-13.
Fusar-Poli, P. Integrated mental health services for the developmental period (0 to 25 years): a critical review of the evidence. Frontiers in Psychiatry, 2019, 10, 355.
62 Please provide a definition of “Latinx” and cite a current peer reviewed reference to support this definition.
Response: I have replaced all instances of “Latinx” with “Hispanic” since the study primarily focuses on a group that speaks Spanish throughout the paper.
75-76 Please delete the Bradley et al., 2008 reference as it can have nothing relevant to provide regarding the COVID-19 pandemic.
Response: I have removed the specified content as suggested.
95-103 These research questions are not well defined. Research questions cannot be compound. As well, they need to be distinct.
Response: I have revised three research questions as suggested. We agree with this comment. Thank you.
“Three research questions guided this study:
- What was the mood state of Latinx adolescents during the COVID-19 pandemic,
and has there been a change in their mood state since the pandemic?
- How do Latinx parents perceive their adolescents’ mood state during the COVID-19 pandemic, and have they noticed any changes in their adolescents’ mood since the pandemic?
- What were the predominant mood states experienced by Latinx adolescents during
the COVID-19 pandemic, and is there a difference in parents’ perceptions of their adolescents’ recent mood changes?”
to
“The following research questions guided this study:
- What were the predominant mood states of Latinx adolescents during the COVID-19 pandemic
- How did Latinx parents perceive their adolescents’ mood states during the COVID-19 pandemic
- Has there been a change in these adolescents’ mood states since the pandemic?
- Have parents noticed any changes in their adolescents’ mood since the pandemic?
- Is there a difference in parents’ perceptions of their adolescents’ recent mood states compared with their perceptions during the pandemic?”
106-118 Please state:
Response: I have provided information regarding points 1-4 below.
- The relationship of the researchers to the school, the parents, and the adolescents
- How informed consent was obtained
- Whether there was an incentive to participate
- How participants were contacted to be involved in this research
- How the survey was designed.
Response: I have provided the details in section 2.2. Measures.
- Any exclusion criteria for participation (whether parents were required to have access to the means to complete online surveys during the school day).
Response: There were no mandatory access requirements for parents, and there were no exclusion criteria for participation.
110 If the questionnaire was sent home, in what way was this an online survey? Please explain in the text.
Response: For the student survey, the school principal scheduled to send the letters to parents explaining the purpose and parent consent form. After that the school principal made announcements to students during school assemblies about the online survey with minor assent form attached. Parent survey was paper form with two language version: English and Spanish with the consent form. The translation was conducted by the native speakers who are qualified to teach Spanish and English both. Student survey was online and English version only. I have explained in the text.
120 Change “Language mostly used at home” to “Most-used language at home”.
Response: Changed.
If N=23, why do, Children in the gifted program, Race/Ethnicity, and Employment status add up to N=22? As well, why does Education level have N=21? Please explain in the text.
Response: The discrepancy arose from missing responses, and I have provided an explanation in the text.
127-130 As well as these citations, please reference studies that have validated each of the index and scales for use during COVID-19.
Response: I have incorporated citations that tested validity and reliability during the COVID-19 period.
Mikkelsen, H. T., Skarstein, S., Helseth, S., Småstuen, M. C., Haraldstad, K., & Rohde, G. Health-related quality of life, health literacy and COVID-19-related worries of 16-to 17-year-old adolescents and parents one year into the pandemic: a cross-sectional study. BMC Public Health, 2022, 22(1), 1321.
Jiang, L., Wang, Y., Zhang, Y., Li, R., Wu, H., Li, C., ... & Tao, Q. The reliability and validity of the center for epidemiologic studies depression scale (CES-D) for Chinese university students. Frontiers in psychiatry, 2019, 10, 315.
140 How do the researchers know that each survey took less than 10 minutes to complete? Please explain in the text.
Response: I have removed the inaccurate information from the content.
141 Change “you/your child spend” to “you/your child spend(s)”.
Response: Changed.
168-168 The graph in the top right corner is the same as the one in the bottom left corner. Only three graphs are needed in this regard, not four.
Response: Removed
171-181 By indicating that over 40% of the adolescents had negative emotions during the pandemic, the authors are not highlighting that the majority of the adolescents had positive emotions. This should be what is highlighted, not the negative.
Response: Thank you for the feedback. I have incorporated the mention of positive emotions as suggested.
172-181 There is no graph provided for either challenge regarding schoolwork or for smartphone usage. Please include these graphs.
Response: I have added graphs.
182 To say that more than half of the adolescents did not feel better after three years of COVID-19 quarantine misrepresents the information provided in the bottom left corner graph found in line 187. The majority felt about the same, very few felt worse or much worse. Please indicate that the majority felt either about the same or better than before the pandemic—as the graph demonstrates.
Response: Thank you for the feedback. I have incorporated the mention of positive emotions as suggested.
184 Figure 2 does not provide any details regarding the breakdown between female and male adolescents. Please provide graphs that indicate this breakdown.
Response: I have included gender information in all the graphs as suggested.
201-205 The bar graphs of the parents have not been prepared to make them comparable to those of the adolescents in the information they provide. Similar to the adolescents, each bar graph should have five bars. Some have only four, or even three. These bar graphs must be redone to provide comparable information to those of the adolescent graphs.
Response: Due to the limited data, some categories had blank cells, resulting in three or four categories, unlike the graphs for children.
191 What in the answers provided by the parents makes the authors conclude that parents focused more on their adolescents' academic-related concerns? Please explain in the text how the authors came to this conclusion.
Response: I have removed the sentence implying that the frequency suggests parents focused more on academics, as it may not be an accurate inference.
232-234 How were the authors able to come to these conclusions regarding the results that were provided to the questions asked of both the adolescents and the parents? There were no questions asking either the adolescents or the parents if they thought their moods were associated or not with the quarantine. These conclusions cannot be drawn as a result.
Response: I have revised the interpretation by eliminating the inference that may not be accurate.
240 As per the template provided to the authors, please separate this section into 4. Discussion, and 5. Conclusion.
Response: I have separated the sections.
241-243 Looking over the results of both the adolescents and the parents, their responses were not that different. Consequently, it is unclear if this study’s findings “align with previous research” or not. Furthermore, the studies referenced here are all pre-COVID-19 and are, therefore, irrelevant to questions concerning COVID-19. The authors need to be precise regarding what can actually be concluded and refer only to other COVID-19 studies.
Response: I have revised the interpretation by eliminating the inference that may not be accurate.
244 The authors says that the change has been “significant”. How does the data from the study demonstrate this? Please explain in the text.
Response: I have removed the section that lacked support from the data in the study.
248,-250 Although previous, non-COVID-19-related research may demonstrate this, how did the results of this study demonstrate that the parents prioritized academic support? Please explain in the text. As well, the research needs to be comparable. This means that other studies referenced must be COVID-19-related.
Response: I have removed the section that lacked support from the data in the study.
250 Again, the authors have focused on the 40% rather than the 60% who did not experience negative mental health. This skews the results inappropriately.
Response: I have revised the content, aiming to achieve a balanced representation of the findings.
258 Merely because the adolescents did not communicate their feelings to their parents does not mean that it was “inadequate”. To make this claim, the authors have to provide research to demonstrate that there is a problem with adolescents not communicating their feelings to their parents in the adolescents having sufficient interventions. The authors found half of the adolescents were part of the gifted program. However, the authors did not breakdown the results regarding the responses of the gifted students and their parents compared with the non-gifted. It is possible that the gifted students did not feel the need to communicate with their parents in this regard because they felt they were taking care of their problems themselves. If making claims about the problem of adolescents communicating with the parents, the authors are asked to breakdown their results between those students enrolled in the gifted program and those not enrolled—both for the results of the adolescents and of their parents.
Response: I replaced “inadequate” into “lack”. And the decision not to break down the results by gifted status was influenced by the small sample size.
265-275 How do the results obtained demonstrate the importance of Latinx adolescents requiring greater attention and support from schools for their emotional well-being? Please indicate the results that lead the authors to this conclusion.
Response: I have removed the section that lacked relevance based on the data in the study.
Furthermore, the authors are asked to reference COVID-19 studies. Only one of the ones referenced in the paragraph is relevant to COVID-19 considerations.
Response: I have included the recent COVID-19 related article.
Hodes, M., & Vostanis, P. Practitioner review: Mental health problems of refugee children and adolescents and their management. Journal of Child Psychology and Psychiatry, 2019, 60(7), 716–731
291-294 This was a study of adolescents and of parents. It was not a study of the role of schools. The authors can draw no conclusions concerning the role that schools might play in this regard as there was nothing in the study conducted that could provide this information. These recommendations must be eliminated.
Response: I have removed the section that lacked relevance based on the data in the study.
294-295 As this study did not consider either teachers or counselors, it cannot provide advice to them. The authors have no information on what teachers or counselors can or cannot do based on the study conducted. These recommendations must be eliminated.
Response: I have removed the section that lacked information from the data in the study.
295-298 There is nothing provided in the results of this study to indicate the necessity of interactive communication between parent and school regarding COVID-19-related mental health of adolescents. As well, the paper referenced is from 2011—well before COVID-19 considerations. This recommendation must be eliminated.
Response: I have removed the section that lacked relevance based on the data in the study.
298-309 Again, the conclusions drawn here go well beyond the scope of what was determined by the study conducted and the research supporting these conclusions are all pre-COVID-19 by many years. The only type of conclusions the authors can draw are those related to the findings of their study, nothing more.
Response: I have removed the section that lacked relevance based on the data in the study.
331 Please check the wording of the question that was on the survey, “During the ongoing Covid-19, how relaxed were lonely?” does not make sense. If this was the wording used on the survey, it must be specified as a limitation of this study.
Response: I have revised as suggested.
352 “What grade are your child in?” is incorrect English. Is this what was on the survey? If so, the problem with the English should be noted. The question should have been either “What grade is your child in?” or “What grades are your children in?”.
Response: I have revised as suggested.
354 “Is your child in gifted program?” is incorrect English. Is this what was on the survey? If so, the problem with the English should be noted. The question should have been “Is your child in the gifted program”.
Response: I have revised as suggested.
355 In Appendix A, line 326, the children were asked if they were “Black” while, in this line, parents were asked if they were “African American”. Please explain in the text why there was a different wording choice for parents—especially because when the data were tabulated there was no mention of “African American”.
Response: I have identified the use of different wording as a limitation of this study.
Reviewer 2 Report
Comments and Suggestions for Authors
1. Please add more keywords (up to 10) and sort them alphabetically.
2. Age of participants were not indicated. This must be addressed.
3. The summed percentage % of females and males does not equal to 100%. I see also some discrepancies with % in other demographic categories. This must be addressed.
4. Lines 125-129: redundancy as this is not related to your study.
5. Section 3.3 is somewhat poor. Please describe clearly (acc. to APA) regression analyses (F-value, p, R-square, etc). Please do not use p < 0.10 as we have replication crisis in psychology and we need strong evidence. Indicate VIF or Tolerance, and Durbin-Watson statistic.
6. Present p-value with 3 decimal places.
7. Add pseudo R-square for logistic regression.
8. Please control at least gender in your regression, and other sociodemographic variables.
9. Table 6. In order to conclude that "Overall, the study found that parents and adolescents had slightly different perceptions of the adolescents’ mood states, as indicated by the correlational analysis (see Table 6)" you need to have at least the same number of participants in each group. But you have 584 adolescents and only 23 parents. Refuse from this table 6 and this analysis, as these are incorrect comparisons. Do not complicate your paper.
10. Indicate N in titles of tables. In general, see APA recommendations for presenting results.
11. Create subsections: Limitations and strengths of the study, Practical Implications, and separate Conclusions.
12. A lack of these data is a very major concern. The Editorial board of the journal should analyse as this is in compliance with the MDPI policies. The study involved adolescents, the ethical approval must be presented.
Institutional Review Board Statement: Not applicable. Why?
Informed Consent Statement: Informed consent was obtained from all subjects involved in this study. From whom? Parents, adolescents? How? Where?
Data Availability Statement: The data is not available publicly. Why?
Author Response
Dear reviewer, I sincerely appreciate your thoughtful review and insightful feedback. I have diligently worked to incorporate your comments and enhance the manuscript comprehensively. I have attached the responses to each comment provided. Thank you once again for your valuable time and consideration!
Response to Review #2 Comments
First of all, thank you so much for taking the time to review this manuscript. Please find the detailed responses below and the corresponding revisions in track changes in the re-submitted files.
- Please add more keywords (up to 10) and sort them alphabetically.
Response: I have added keywords as suggested.
- Age of participants were not indicated. This must be addressed.
Response: I have included the age of participants.
- The summed percentage % of females and males does not equal to 100%. I see also some discrepancies with % in other demographic categories. This must be addressed.
Response: The discrepancy arose from missing responses, and I have provided an explanation in the text.
- Lines 125-129: redundancy as this is not related to your study.
Response: I have revised accordingly as suggested.
- Section 3.3 is somewhat poor. Please describe clearly (acc. to APA) regression analyses (F-value, p, R-square, etc). Please do not use p < 0.10 as we have replication crisis in psychology and we need strong evidence. Indicate VIF or Tolerance, and Durbin-Watson statistic.
Response: I have revised the table by adding F, P (three decimals), and R-square in the table. Additionally, I have included VIF and Durbin-Watson statistic accordingly.
- Present p-value with 3 decimal places.
Response: I have revised accordingly.
- Add pseudo R-square for logistic regression.
Response: Cox & Snell R-Square values are reported instead of pseudo R-square, as they are considered equivalent measures in logistic regression. SPSS does not report pseudo R-square. Thank you.
- Please control at least gender in your regression, and other sociodemographic variables.
Response: Gender was treated as a control variable.
- Table 6. In order to conclude that "Overall, the study found that parents and adolescents had slightly different perceptions of the adolescents’ mood states, as indicated by the correlational analysis (see Table 6)" you need to have at least the same number of participants in each group. But you have 584 adolescents and only 23 parents. Refuse from this table 6 and this analysis, as these are incorrect comparisons. Do not complicate your paper.
Response: Thank you for the feedback. We agree with the comment. Thus, we have removed the entire section that was deemed unreliable due to the extremely unbalanced sample size between groups.
- Indicate N in titles of tables. In general, see APA recommendations for presenting results.
Response: I have indicated as requested.
- Create subsections: Limitations and strengths of the study, Practical Implications, and separate Conclusions.
Response: I have separated sections.
- A lack of these data is a very major concern. The Editorial board of the journal should analyse as this is in compliance with the MDPI policies. The study involved adolescents, the ethical approval must be presented.
Response: I have revised sections below.
Institutional Review Board Statement: Not applicable. Why?
Response: The study was approved from both the Institutional Review Board of Kennesaw State University and Dalton Public schools in Georgia, U.S..
Informed Consent Statement: Informed consent was obtained from all subjects involved in this study. From whom? Parents, adolescents? How? Where?
Response: For the student survey, the school principal arranged to dispatch letters to parents explaining the purpose and including the parent consent form. Following that, the school principal made announcements to students during school assemblies about the online survey, with a minor assent form attached.
Data Availability Statement: The data is not available publicly.
Response: The data presented in this study are available on request from the corresponding author
Round 2
Reviewer 1 Report
Comments and Suggestions for Authors
This version is improved from the first submission. However, the authors are still holding on to their view that COVID-19 caused mental health difficulties for Hispanic adolescents in disproportion to other adolescents and that their mental health difficulties were more relevant to discuss than the actual finding that the majority of the Hispanic adolescents surveyed experienced no negative mental health. Furthermore, of the few parents that responded, the majority had views in line with their children’s views of themselves. In contrast, the authors are wanting to stress that because the adolescents didn’t share their views with their parents that this was in itself a problem. They do this because the older researcher they are referencing indicates that adolescents with mental health difficulties do better when they talk with their parents. However, there is nothing that was found in the surveys undertaken by the authors to demonstrate that those who were having mental health concerns were the ones who were not talking with their parents as the authors did not provide the evidence that it was these adolescents in particular who were experiencing the greatest problems with their mental health.
The authors must set aside their bias and look again at their research findings to rewrite their paper to focus on the positive finding that the majority of the adolescents had no additional problems regarding their mental health as a result of COVID-19. Although the authors do note this—and do so at the beginning of their interpretations of their data—this is not sufficient. The paper remains skewed towards the minority finding that some adolescents did experience negative mental health.
Please check the template for Healthcare regarding references. Note that references should be numbered consecutively, beginning with “1”. They are not to be numbered in relation to their alphabetical appearance. Please redo the citations so that they are number from “1” onwards. As well, redo the ordering of the references so that they are ordered by how they appear in the text, not alphabetically.
The authors have supplemented their references with a number of COVID-19-related publications. Thank you. This is what is expected. However, the references list is still primarily featuring pre-COVID-19 material. For a paper published regarding COVID-19, all references should be published from 2020 onwards unless the paper is historical, comparing COVID-19 with previous pandemics (which this paper does not). Please eliminate the dependence on pre-COVID-19-related references and eliminate them from the reference list. What would be best is if the authors replaced these with COVID-19-related references.
Line by line suggested edits:
17 Change “insufficient of understanding” to “insufficient understanding”.
28 Reference 31 regards Asian Americans from 2009—it is completely inappropriate to reference. Please delete. Reference 53 concerns obesity, not pandemics—please delete.
32 Change “only one third” to “few”
33 Thank you to the authors for adding citations 16 and 17. Please now delete 3, 12, and 42, which are much too old.
34-35 None of the studies cited are since COVID-19. Please delete all of them of find studies that were published since 2020 to support these claims.
40 None of the studies cited are since COVID-19. Please delete all of them of find studies that were published since 2020 to support these claims.
43 Only citation 28 is to research done since the beginning of COVID-19. The other two references are too old. Please delete them and find references published since 2020 to support the claims.
50-52 This is an important claim for this paper, yet the authors have provided no evidence that studies have found this to be true. Please cite a reference published since COVID-19 to support this claim.
61 Reference 1 concerns Pakistan. Reference 11 relates to Malaysia. Neither are studies concerning Hispanic parents. These cannot be cited as supporting references for the claim. Please substitute studies regarding Hispanic parents.
67 Although it is fine to cite reference 32 as it was published in 2020, the authors are writing in 2023 and are using citation 32 to support current mental health problems of children and adolescents. The authors must find a 2023 reference that continues to support this drastic increase. Otherwise, the authors need to note that this drastic increase was seen at the beginning of COVID-19 and has changed since then.
69-71 Since quarantine restrictions and social distancing requirements are now over, the authors need to specify the time period these were in effect and not appear to be stating that they are continuing today.
75 Reference 36 is from a study done in April and May of 2020, at the beginning of the pandemic. The authors need to find more recent evidence that their claims persisted throughout the pandemic and have left lasting results.
90 As mentioned in the previous review, research questions must be distinct and not include compound questions. There cannot be an “and” in a research question. Research question 1 must be divided in to two research questions.
93 The same problem exists with research question 2 as in research question 1. This was mentioned in the previous review. There cannot be an “and” in a research question. Please divide research question 2 into two questions.
104 If the authors sent home paper versions of the survey, how do the authors know that the parents completed the survey during school hours? Please explain this in the text.
121-122 If the authors are calling the one heading “Frequency” rather than “Number”, the column to the left of this must be labeled “Value”. Please also label the first column on the left as “Division” or “Category”.
122 Change “ btween” to “between”.
138 A COVID-19-related reference will be needed to support citation 46.
144 Why did not all of the responses use a 5-point Likert scale? Please explain in the text.
149-150 Please explain in the text why the authors chose to use “During the ongoing COVID-19 pandemic, have you/your child cried often?” as the question asked on the survey rather than “During the ongoing COVID-19 pandemic, how often did you/your child cry”? The question following should read “Compared to during the ongoing COVID-19 pandemic and now, how would you/your child the general feeling now?”
154 Item 11 on which survey—the adolescent or the parent? Please answer this in the text.
156 Change “Items 4 to 9 and 12 to 14” to “Items 4 to 9 on the adolescent survey and 12 to 14 on the parent survey”.
163 Item 11 on which survey—the adolescent or the parent? Please answer this in the text.
166 Please explain in the text why SPSS v. 28 was used for the analysis and provide a COVID-19-related reference demonstrating that it was used for a similar COVID-19 study.
219-220 It needs to be explained in the text why “Never” is not recorded in the first graph or else add a new bar for “Never” with a frequency of “0”.
220-221 It needs to be explained in the text why “Always” is not recorded in the first graph or else add a new bar for “Always” with a frequency of “0”. Furthermore, it needs to be explained in the text why neither “Always” nor “Most of the time” were not recorded in the second graph or else add a new bar for “Always” with a frequency of “0” and a new bar for “Most of the time” with a frequency of “0”.
221-222 It needs to be explained in the text why “Agree” is not recorded in the second graph or else add a new bar for “Agree” with a frequency of “0”.
222-223 It needs to be explained in the text why “0-1 hour” is not recorded in the first graph or else add a new bar for “0-1 hour” with a frequency of “0”. Furthermore, it needs to be explained in the text why “Much worse now” was not recorded in the second graph or else add a new bar for “Much worse now” with a frequency of “0”.
258 Please include a more recent reference to support citation 57.
260 “suggestive of depression or anxiety”—Please include a current reference to support this claim.
260-262 Please include a current reference to support this claim.
266 Both of these citations are to research that is too old to be relevant to COVID-19. Please provide COVID-19-related references.
267 Change “Limitations and strength of the Study” to “Limitations”.
279-286 In the conclusion, the authors have missed pointing out that the majority of Hispanic adolescents did not experience any mental health changes related to COVID-19 with respect to the questions asked on the survey. As well, their parents responses were similar in this regard. The authors must point out that their advice pertains only to the minority of students who did have mental health challenges.
285-286 These citations are all irrelevant as none of them pertain to the COVID-19 situation and they are all to older research.
287-295 The authors cannot present these as practical implications as they did not study schools, local communities and different levels of government. Please delete this entire section.
299-300 As there are only two others, who are these “others”? Please be specific in the text.
340 Change “grades are” to “grade is”.
Comments on the Quality of English LanguageAll comments on the quality of English Lanuage use are stated in the Comments and Sugestions for Authors.
Reviewer 2 Report
Comments and Suggestions for Authors
The paper was improved satisfactprily. Minor comments:
1. Long paragraphs are unwanted.
2. P-value can not equal 0.000, plese use p < 0.001. See table 5.
3. Please do not use "on the other hand" if you have no "on the one hand" before.
4. Table 4. Indicate in the table title what you are predicting. What is your dependen variable? XXX....predicting smth.
Author Response
Response to Review #2 Comments
First of all, thank you so much for taking the time to review this manuscript again. Please find the detailed responses below and the corresponding revisions in track changes in the re-submitted files.
The paper was improved satisfactory. Minor comments:
Long paragraphs are unwanted.
Response: I have attempted to condense it. It was not easy, but I appreciate your understanding.
P-value can not equal 0.000, plese use p < 0.001. See table 5.
Response: I have revised it accordingly. Thank you.
- Please do not use "on the other hand" if you have no "on the one hand" before.
Response: I have changed it to “conversely”.
- Table 4. Indicate in the table title what you are predicting. What is your dependen variable? XXX....predicting smth.
Response: The dependent variable is specified at the bottom of Table 4.
DV: Current mood change